# A new power-law model for $\mu$-$\Lambda$ relationships in convective and stratiform rainfall

Christos Gatidis[1], Marc Schleiss[1], and Christine Unal[1]

[1]Department of Geoscience and Remote Sensing, Delft University of Technology, Delft, The Netherlands

**Correspondence:** Christos Gatidis (C.Gatidis@tudelft.nl)

**Abstract.** In this study, we take a closer look at the important issue of $\mu$-$\Lambda$ relationships in raindrop size distributions (DSD) by conducting a systematic analysis of twenty months of data collected by disdrometers in the Netherlands. A new power-law model for representing $\mu$-$\Lambda$ relationships based on double normalization framework is proposed and used to derive separate $\mu$-$\Lambda$ relationships for stratiform and convective rain events. The sensitivity of the obtained relationships to measurement uncertainty is studied by applying two different quality control filters based on the mass-weighted mean drop diameter ($D_m$) and liquid water content (LWC). Our results show that there are significant differences in $\mu$-$\Lambda$ relationships between convective and stratiform rainfall types. However, the retrieved relationships appear to be quite robust to measurement noise and there is a good agreement with other reference relations for similar climatological conditions.

# 1   Introduction

The $\mu$-$\Lambda$ relationship in rainfall micro-physics refers to a deterministic function linking the shape ($\mu$) and scale ($\Lambda$) parameters of a gamma raindrop size distribution (DSD) model (Zhang et al., 2001). Such relationships are important for understanding the microstructure and dynamics of precipitation and are essential for retrieving DSDs from polarimetric radar measurements. The primary use of $\mu$-$\Lambda$ relationships in radar remote sensing is to reduce the number of model parameters (from three to two) in DSD retrieval algorithms. However, DSD retrieval remains challenging and subject to various sources of uncertainty,

including the accuracy of the remote sensing observations, the limitations of the DSD retrieval algorithms and the choice of the $\mu$-$\Lambda$ relationship.

Numerous $\mu$-$\Lambda$ relationships have been proposed in the literature, with second-order polynomial functions being the most popular. The first relationships were proposed by Zhang et al. (2001, 2003) using DSD data collected in Florida, USA. Since then, several other relationships have been proposed for different datasets and rainfall climatologies. For example, van Leth

et al. (2020) derived a relationship for the Netherlands using nine months of disdrometer data in Wageningen. Their relationship differs from those reported by Zhang et al. (2001, 2003) which is reasonable given that stratiform rain dominates in the Netherlands and convective and stratiform precipitation have different DSDs. Notably, the drop sizes in convective rain tend to be larger and more variable, which results in a broader DSD with smaller $\mu$ and $\Lambda$ values. Conversely, raindrops in stratiform rain are typically smaller and more uniform in size, corresponding to larger $\mu$ values for a given $\Lambda$. Vivekanandan et al. (2004)

pointed out that correlation between $\mu$ and $\Lambda$ exists but may vary across different types of rain, highlighting the need for further understanding of $\mu$-$\Lambda$ variability. Despite the fact that the $\mu$-$\Lambda$ relationship changes depending on rain-type, Chu and Su (2008) has shown that $\mu$-$\Lambda$ relations exhibit similar behaviour for small $\mu$ values, which usually correspond to heavier rainfall events, while the relations start to deviate as $\mu$ and $\Lambda$ increase, indicating light to moderate rain events.

At the microphysics scale, Bringi et al. (2003) showed that a linear relationship with a negative slope exists between the

generalized intercept parameter ($N_w$) in logarithmic scale and the mass-weighted mean diameter ($D_m$) for stratiform rainfall. For convective rain, two clusters of data emerge, with one cluster consisting of maritime-like convective points and the other of continental-like points. The latter is characterized by larger raindrop sizes and lower concentration, whereas the former exhibits the opposite trend, with a higher concentration of smaller-sized drops.

Similarly, other studies have examined discrepancies in $\mu$-$\Lambda$ relationships based on either regional (Chen et al., 2016) or

seasonal criteria (Seela et al., 2018), showing that both factors are influenced by the prevailing climatic conditions and the dominant rain type. Besides the rain type and climatology, other factors that could potentially affect $\mu$-$\Lambda$ relation have also been partially investigated, such as sampling errors (Zhang et al., 2003), temporal sampling resolution, and the adequacy of the gamma model itself (Gatidis et al., 2022). Zhang et al. (2003) discussed how sampling errors or deviations from the gamma distribution could result in a correlation between $\mu$ and $\Lambda$. Using DSD observations of moderate-intensity, stratiform rain events

in the Netherlands, Gatidis et al. (2022) found that the $\mu$-$\Lambda$ relationship remained robust regardless of the sampling resolution and the validity of the gamma model.

Another issue that arises when studying $\mu$-$\Lambda$ relationships is the rainfall classification. Several techniques have been proposed to classify rainfall into stratiform and convective regimes using a variety of different sensors. These methods may include weather radar data, Micro Rain Radar (MRR) vertical profiles and machine learning models for the bright band detection (Ghada et al., 2022; Romatschke and Dixon, 2022; Qi et al., 2013; Powell et al., 2016). For example, Yang et al. (2019) used a K-nearest neighbor supervised machine learning algorithm for the classification and Doppler radar data to train the model. Other studies use a combination of ground based sensors like rain gauges or disdrometers and radar data (Ulbrich and Atlas, 2007; Tokay and Short, 1996; Bringi et al., 2003). In this work, the stratiform/convective classification relies primarily on rain intensity estimations by disdrometer, data from a cloud radar and vertical profiles of reflectivity from a MRR for detecting the melting layer. Additionally, a combination of CAPE and lightning activity data assists in making the final classification decision.

In this paper, we take a closer look at $\mu$-$\Lambda$ relationships for convective and stratiform rain. Twenty months of DSD data were collected in the Netherlands using two co-located Parsivel$^2$ optical disdrometers. Our analysis starts by applying a quality control filter on $D_m$ and LWC to discard observations for which the two sensors showed large disagreement. Within the double-moment normalization framework, a new $\mu$-$\Lambda$ power-law relationship is introduced and fitted to the remaining data, resulting in coefficients with meaningful physical interpretation. Finally, the data are classified into convective and stratiform rain and differences between the derived $\mu$-$\Lambda$ relationships are highlighted.

The work is organized as follows. In Section 2, we introduce the data used, and in Section 3 the methodology is presented. In Section 4, the main results for the quality control filter and the $\mu$-$\Lambda$ relationship analysis for the different rainfall regimes are shown. Finally, the conclusions are provided in Section 5.

## 2  Data

The DSD data used in this study were collected by two co-located, perpendicularly oriented Parsivel$^2$ (Particle Size and Velocity) optical disdrometers (hereafter Parsivel 1 and Parsivel 2) in Cabauw, a polder area located in the western part of the Netherlands between January $1^{st}$ 2021 and August $31^{st}$ 2022. The disdrometer data were collected within the framework of the Ruisdael Observatory, a national research infrastructure that consists of a large network of observations and models in the Netherlands where data are merged together to study atmospheric processes across scales and achieve a better understanding of climate change and weather (Russchenberg et al., 2022). The measurement principle and characteristics of the Parsivel$^2$ have already been extensively described in previous studies (Löffler-Mang and Joss, 2000; Thurai et al., 2011; Tokay et al., 2014) and will not be repeated here. In the past, several studies have highlighted the effect of strong winds on Parsivel observations (Friedrich et al., 2013a; Lin et al., 2021), which could result in unrealistic big raindrops with small fall velocities. Thus, Friedrich et al. (2013b) proposed a quality control method for removing all these spurious observations. In present work even though no action was taken in this direction, the observations from the two co-located sensors were compared to each other. Whenever the agreement between the two sensors was low, the DSDs were removed from the analysis. The total dataset used for this study consisted of 21,178 1-minute DSDs. After filtering, the dataset was reduced to 16,975 DSDs. A detailed description

of the filtering process will be given in a following section. No effort was made to investigate the reasons behind the occasional disagreements. The latter have already been extensively studied and documented in the literature and include, among other, errors due to wind, sampling, splashing and internal processing.

In addition to the disdrometer data, the following resources were used for visualisation purposes and qualitative precipitation classification:

- Radar data collected by CLARA (CLoud Atmospheric RAdar), a dual-frequency (35-94 GHz) polarimetric scanning cloud radar in Cabauw (https://cloudnet.fmi.fi/search/data?site=cabauw).

- Vertical profiles of reflectivity from a MRR at Cabauw (https://dataplatform.knmi.nl/dataset/ruisdael-mrr-cabauw-2).

- Convective available potential energy from ERA5, ECMWF reanalysis data, (https://doi.org/10.24381/cds.adbb2d47).

- Lightning activity (strikes) from the ZEUS long-range cloud-to-ground lightning detection system
(https://www.meteo.gr/talos/en/).

## 3 Methodology

The methodology can be summarized as follows. Firstly, rain events are classified into two types: convective and stratiform. The data from the two co-located disdrometers are then used to fit a gamma model for each 1-min time interval and derive the corresponding shape ($\mu$) and slope ($\Lambda$) parameters. The data from the two disdrometers are cross-checked and any time steps
for which the two sensors disagree with each other are removed. The remaining data are used to fit the overall $\mu$-$\Lambda$ relation, as well as the relations for convective and stratiform rainfall types. Finally, the results are compared with those available in the literature to ensure consistency and validity.

### 3.1 DSD model / Parameter fitting

The DSD N(D) ($mm^{-1}$ $m^{-3}$) is modeled using a normalized gamma distribution with shape parameter $\mu$ (-), slope $\Lambda$ ($mm^{-1}$)
and intercept $N_w$ ($mm^{-1}$ $m^{-3}$) as in (Bringi et al., 2003; Testud et al., 2001):

$$N(D) = N_w f(\mu) \left( \frac{D}{D_m} \right)^{\mu} e^{-(4+\mu)\frac{D}{D_m}} \tag{1}$$

$$f(\mu) = \frac{6}{4^4} \frac{(\mu+4)^{(\mu+4)}}{\Gamma(\mu+4)}, \tag{2}$$

$$N_w = \frac{4^4}{\pi \rho_w} \left( \frac{LWC}{D_m^4} \right), \tag{3}$$

$$D_m = \frac{\int_{D_{min}}^{D_{max}} N(D)D^4 dD}{\int_{D_{min}}^{D_{max}} N(D)D^3 dD} = \frac{4+\mu}{\Lambda}, \tag{4}$$

$$LWC = \frac{\pi \rho_w}{6} \int_{D_{min}}^{D_{max}} N(D)D^3 dD. \tag{5}$$

In the equations above, $D_m$ (mm) is the mass-weighted mean diameter, LWC (g m$^{-3}$) the liquid water content, $\rho_w$ ($10^{-3}$ g mm$^{-3}$) the density of liquid water and $D_{min}$-$D_{max}$ are the integration limits due to the finite range of drop sizes which can occur in nature. This model has been extensively used and assessed in the literature (Gatidis et al., 2020; Thurai et al., 2019). Similarly to Bringi and Chandrasekar (2001); Gatidis et al. (2020); Thurai et al. (2014), the method of moments and more particularly the $3^{rd}$ and $4^{th}$ DSD moments were used to fit the gamma DSD and estimate the three unknown parameters $\mu$, $\Lambda$ and $N_w$ from empirical DSD spectra, with $\mu$ values ranging between -3 and 15, as described by Thurai et al. (2014). The advantages and disadvantages of method of moments with respect to other methods such as maximum likelihood estimation were discussed in previous studies (Smith and Kliche, 2005; Smith et al., 2009; Kliche et al., 2008; Gatidis et al., 2020) and will not be repeated here.

## 3.2   $\mu$-$\Lambda$ relationship

Numerous empirical $\mu - \Lambda$ relationships have been proposed and discussed in the literature (Zhang et al., 2003; van Leth et al., 2020; Gatidis et al., 2022). The most common is the second-order polynomial model proposed by Zhang et al. (2001):

$$\mu = -0.016\Lambda^2 + 1.213\Lambda - 1.957. \tag{6}$$

While polynomial relationships are a practical way to represent empirical $\mu - \Lambda$ relationships, they lack theoretical justification, and their coefficients do not have clear physical interpretations. Thus, we propose an alternative model that offers better justification and interpretation. Our model is:

$$\Lambda = \alpha(\mu+3)^\beta(\mu+4)^{1-\beta}, \tag{7}$$

where $\alpha$ (mm$^{-1}$) and $\beta$ (-) are two model coefficients inferred using a non-linear least squares fit on pairs of ($\mu$,$\Lambda$) values.

Justification:

The $\mu - \Lambda$ relationship in Eq. (7) can be derived from the double-moment normalization framework by Lee et al. (2004). In this

framework, the DSD is expressed as $N(D) = N_c h(\frac{D}{D_c})$ where $D_c$ (mm) is a characteristic drop diameter that depends on two references moments, $N_c$ (mm$^{-1}$ m$^{-3}$) is a drop number concentration parameter and $h$ a template function for describing the shape of the normalized DSD. The two reference moments $M_i$ and $M_j$ used for the normalization depend on the application. In all generality:

$$D_c = \left(\frac{M_j}{M_i}\right)^{\frac{1}{j-i}},$$ (8)

$$N_c = M_i^{(j+1)(j-i)} M_j^{(i+1)(i-j)}.$$ (9)

To simplify, we consider the special case in which $j = i+1$ and $D_c = M_j/M_{j-1}$. For example, when j=4 and i=3, we get $D_c$ = $M_4/M_3 = D_m$. If in addition we assume that the DSD is gamma, then we get the model for N(D) as in Eq. (1).

One key property of the double-moment normalization framework is that any moment $M_n$ of the DSD can be expressed as a power law of the characteristic drop size $D_c$:

$$M_n = \int_0^\infty D^n N(D) dD = N_c \xi_n D_c^{n+1},$$ (10)

where

$$\xi_n = \int_0^\infty x^n h(x) dx.$$ (11)

However, since the DSD variability might not be fully captured by two reference moments, we will assume that:

$$M_n = N_c a_n D_c^{b_n},$$ (12)

where $a_n$ and $b_n$ are two empirical coefficients which can be slightly different from their theoretical expressions in Eq. (10). Assuming Eq. (12) holds, we must have:

$$\frac{M_n}{M_{n-1}} = \frac{a_n}{a_{n-1}} D_c^{b_n - b_{n-1}}.$$ (13)

Considering that the DSD is assumed to follow a gamma model, and given that $\int_0^\infty D^a e^{-bD} dD = \Gamma(a+1)/b^{(a+1)}$ and $\Gamma(a+1) = a\Gamma(a)$, where $\Gamma(a)$ is gamma function, then $D_c$ (the ratio of two successive reference moments with $i = j-1$) is given by:

$$D_c = \frac{M_j}{M_{j-1}} = \frac{\mu+j}{\Lambda}.$$ (14)

Combining Eq. (13) and Eq. (14) yields:

$$\frac{M_n}{M_{n-1}} = \frac{a_n}{a_{n-1}} \left( \frac{\mu+j}{\Lambda} \right)^{b_n - b_{n-1}}. \tag{15}$$

For a gamma DSD, the left hand side is: $\frac{\mu+n}{\Lambda}$. Therefore,

$$\frac{\mu+n}{\Lambda} = \frac{a_n}{a_{n-1}} \left( \frac{\mu+j}{\Lambda} \right)^{b_n - b_{n-1}}, \tag{16}$$

which can be rewritten as:

$$\Lambda = \alpha_n (\mu+n)^{\beta_n} (\mu+j)^{1-\beta_n}, \tag{17}$$

where $\beta_n = (b_{n-1} - b_n + 1)^{-1}$ and $\alpha_n = \left( \frac{a_{n-1}}{a_n} \right)^{\beta_n}$. This leads to a general $\mu - \Lambda$ relationship of the form:

$$\Lambda = \alpha (\mu+n)^{\beta} (\mu+j)^{1-\beta}, \tag{18}$$

where $\alpha$ and $\beta$ depend on the two chosen pairs of consecutive reference moments $(M_{j-1}, M_j)$ and $(M_{n-1}, M_n)$. In particular, if n=3 and j=4, then we get $D_c = D_m$ and Eq. (7), which is the equation we will use in this study. Note that the choice n=j is impossible because it just leads to a self-consistency constraint $D_c = \frac{\mu+4}{\Lambda}$. In other words, for any characteristic drop size $D_c$, two additional moments are needed to estimate the scaling law linking $M_n$ to $D_c$.

Eq. (18) is interesting because it shows that within the framework of double-moment normalization, the relationship between $\mu$ and $\Lambda$ depends on the chosen reference moments used to fit and/or model the DSD. This is a finding that had been previously hinted by other studies, such as Seifert (2005), but had not been fully explained until now.

## 3.3  DSD filtering

One advantage of having co-located disdrometers is that the DSD measurements can be cross-checked to make sure they are consistent with each other. Suspicious DSDs are identified in a two-step procedure: First, the $D_m$ values for both disdrometers are calculated from the measured DSD spectra. If the absolute value of the difference in $D_m$ values for two co-located measurements exceeds 0.5 mm, both DSD spectra are discarded. The 0.5 mm threshold is inspired by the Global Precipitation Measurement (GPM) mission which states that $D_m$ should be known to within $\pm$ 0.5 mm Tokay et al. (2020). Then, a second filter that uses a relative-error threshold of $\pm$ 50% on the LWC between Parsivel 1 and Parsivel 2 is applied. The justification for this second filter can be found in Eq. (3) which shows the linear relation between $N_w$ and LWC (assuming $D_m$ is known). The use of a relative error threshold means that the DSDs corresponding to low values of LWC (i.e., low rainfall intensities) are filtered more strictly than the DSDs corresponding to moderate and high values of LWC.

## 3.4 Stratiform/Convective classification

In the literature, various methods have been introduced for rain type classification, utilizing different datasets and techniques. One popular method referred to as BR03 (Bringi et al., 2003) based on disdrometer data, uses the standard deviation of the rain rate over a 10-minute moving time window. If the standard deviation exceeds 1.5 mm h$^{-1}$, the period is classified as convective; otherwise, it is labeled as stratiform. Figs. 1 and 2 illustrate the application of the BR03 method to our cloud radar and disdrometer data respectively, collected in Cabauw on May 22$^{nd}$, 2021, during a 3-hour period of stratiform rain. The BR03 method identified two short convective periods within the event. However, the 35 GHz cloud radar co-polar correlation coefficient reveals a distinct melting layer signature throughout the entire event, which contradicts the classification suggested by BR03.

To avoid issues with an automated procedure for rain type classification, we manually classified each time period based on the available data sources. To be classified as convective, a time period had to meet the following criteria:

1. Rainfall intensity (by disdrometer) above 10 mm h$^{-1}$.

2. No melting layer signature in the cloud radar and MRR.

3. Convective available potential energy (CAPE) above 1000 J/kg.

4. Lightning activity around Cabauw.

To determine the convective events, we start by identifying all 1-min DSD measurements for which the rain rate exceeds 10 mm h$^{-1}$. We then remove all periods for which there is a clear melting layer signature, since these correspond to stratiform rain. Regarding requirements 3 and 4, please note that no processing was performed on the associated data sets. CAPE and lightning activity are only used as additional diagnostic variables to help with the final classification decision. For the final selection of convective events, only the periods for which the CAPE values were larger than 1000 J/kg and for which lightning strikes were detected over the Cabauw area are kept. High CAPE level indicate favourable conditions for strong updrafts and storm development, potentially leading to convective rain, while lightning is a phenomenon that can accompany convective storms. However, it is important to state that they are not the exclusive drivers of convective processes (Schumacher et al., 2013). In this study, they are used as an additional indicator for potential convection which together with the high rain intensity and the absence of melting layer will ensure that no false convective events are identified. The reasoning behind this approach is that we think it is preferable to be too strict and exclude a few convective events rather than being too tolerant and include some stratiform or mixed-type events into the convective dataset.

Table 1 presents an overview of the eight convective events that were identified in this way, together with some basic statistics for R, N$_w$, D$_m$ and LWC. All eight convective events occurred during late spring and summer and were associated with moist unstable atmospheric conditions (i.e., thermal convection). The average rainfall intensity for the convective events is between 15.1 and 123.1 mm h$^{-1}$ and the highest intensity occurred on 19 May 2022 (mean LWC of 6.1 g mm$^{-3}$ and average D$_m$ of 2.4 mm).

Note that while we are confident that all our convective events were indeed convective, it is likely that some additional cases of convective rainfall were missed and wrongly attributed to the stratiform case because they did not meet all of the requirements mentioned above. However, since the Netherlands experiences predominantly stratiform rainfall, the inclusion of a few convective cases in the stratiform category is likely to have minimal impact on the results.

## 4   Results

### 4.1   Quality control of DSD data

For the quality control of the DSD data, initially the $D_m$ filter is applied as was described in Section 3.3. This first filter substantially reduces the measurement uncertainty affecting the $D_m$ values. The root mean square difference (RMSD) on measured $D_m$ values decreases from 0.32 mm to 0.14 mm and the Pearson correlation coefficient increases from 0.53 to 0.88. However, the scatter on $\log_{10}(N_w)$ is still high (RMSD of 0.32 and correlation of 0.70).

Therefore, the second filter on LWC values is used. Fig. 3 shows the $N_w$ values in logarithmic scale before and after the two filters on $D_m$ and LWC. We can see that the combination of these two filters greatly reduces the scatter. The correlation coefficient increases from 0.70 to 0.86 and the RMSD decreases from 0.32 to 0.16. The LWC filter also slightly improves the agreement on $D_m$ (correlation coefficient increases from 0.88 to 0.90 and RMSD reduces from 0.14 to 0.12 mm). In total, 19.8% of the DSDs were discarded during the filtering.

### 4.2   Fitted $\mu$-$\Lambda$ relationships

First, the overall $\mu$-$\Lambda$ relationship without any distinction for the rainfall type is presented. For this part, all 1-min pairs of $(\mu,\Lambda)$ values from the two disdrometers were combined into a single dataset and the optimal $\alpha$ and $\beta$ coefficients of the power-law in Eq. (7) were fitted using non-linear least squares. To assess the effect of the quality control procedure, the analysis was done with and without the $D_m$-LWC filters. However, to our surprise, the optimal power-law coefficients ($\alpha$=1.632 and $\beta$=5.038) of

the $\mu - \Lambda$ relationship with/without filters were almost the same. Similarly, the RMSD values and goodness of fit with/without filters were identical. The results above are highly encouraging, as they suggest that the suspicious DSDs removed during quality control were mainly affected by random noise rather than systematic errors. Consequently, the filters applied did not significantly impact the overall $\mu - \Lambda$ relationship, except for reducing the measurement uncertainty. Furthermore, the $\mu - \Lambda$ relationship for each disdrometer was obtained and then compared. There is a relatively good agreement between the two

sensors, particularly for smaller $\mu$ values ($\mu < 4$) where RMSD of $\Lambda$ values is 0.28 mm$^{-1}$. For cases with $\mu$ greater than 4, the RMSD increases to 1.1 mm$^{-1}$. The slightly bigger differences between the two relations for higher $\mu$ values can be explained by the existing sampling uncertainty in the lower rainfall intensities. All the above imply that a single disdrometer may suffice to derive representative $\mu$-$\Lambda$ relationships without requiring co-location.

    Next, the stratiform-convective classification procedure as described in Section 3.4 was applied. Note that for this part of

the analysis, only the DSD measurements that passed the $D_m$-LWC filters were used. The obtained $\mu$-$\Lambda$ relationships for each

rainfall type are presented in Fig. 4. We can see that there are two clearly different $\mu - \Lambda$ relationships for the stratiform and convective rain events. Although the DSD data for the convective regime originate from eight distinct events, the $(\mu,\Lambda)$ pairs corresponding to them nicely align with each other along the fitted power law. This is remarkable given that the $\mu$ values cover a relatively large range from -1 to 9. However, it should be highlighted that predictions for $\mu > 9$ in convective events should be interpreted very carefully, given that we do not have any observations beyond this range. The data for the stratiform cases also nicely follow the power-law model, albeit with larger scatter. The $\mu$ values corresponding to the stratiform cases also cover a larger range of values from -2 up to 15, with the most probable value being between 2 and 6. Note that $\mu$ values exceeding 15 are possible but only the DSDs with $\mu < 15$ were used in this study.

The stratiform relationship shows striking similarity to the results obtained by van Leth et al. (2020) and Gatidis et al. (2020) who also focused on stratiform rain in the Netherlands with low to moderate rainfall intensities. Compared to the convective one, the stratiform relationship predicts higher $\Lambda$ values for a given $\mu$, which is consistent with lower $D_m$ values. The convective $\mu$-$\Lambda$ relationship is similar to the ones obtained by Zhang et al. (2001, 2003) in Florida during the summer months in an environment that is prone to convection due to thermal instability and tropical cyclones. It is worth noting that for small $\mu$ values ranging from -2 to 4, corresponding to higher rainfall rates, the stratiform and convective relationships exhibit remarkable similarity, reflected in a RMSD of 0.77 mm$^{-1}$ for $\Lambda$ values. For $\mu$ values greater than 4, larger deviations between the two relationships can be noted (RMSD = 4.96 mm$^{-1}$). The fact that the two relationships diverge for higher $\mu$ values can be attributed to the fact that the characteristic drop sizes for a given DSD shape tends to be higher for convective events, which becomes more visible when the DSDs are peaked (i.e., large $\mu$). The fact that the Parsivel struggles to detect small raindrops is unlikely to explain the differences since all suspicious DSDs for which the two co-located disdrometers disagreed with each other were removed prior to analysis.

The significant differences we see between convective and stratiform $\mu$-$\Lambda$ relationships suggests that choosing a good relationship is key for retrieving physically meaningful and realistic DSDs from polarimetric radar observations, even though the exact consequences of a wrong $\mu$-$\Lambda$ relation to the DSD retrieval procedure still requires further investigation. Using a single, global $\mu$-$\Lambda$ relationship regardless of the rainfall type could be problematic, especially for lower rainfall rates and very peaked DSDs.

## 5   Conclusions

A study was conducted to analyze $\mu$-$\Lambda$ relationships in convective and stratiform rainfall in the Netherlands. Twenty months of DSD data were collected in Cabauw using two co-located Parsivel[2] optical disdrometers. A quality control filter on $D_m$ and LWC was applied to eliminate periods during which the two disdrometers showed large disagreement. Subsequently, the data from both sensors were combined, and a new $\mu$-$\Lambda$ power-law relationship based on the double-moment normalization framework was fitted. According to the results the following conclusions can be drawn.

1. The $D_m$-LWC filter based on two co-located disdrometers substantially reduces the uncertainty affecting the measured DSDs but does not change the $\mu$-$\Lambda$ relationship. This means that reliable $\mu$-$\Lambda$ relationships can be obtained using a single disdrometer.

2. The $\mu$-$\Lambda$ relationships differ significantly between convective and stratiform precipitation, particularly for higher $\mu$ and $\Lambda$ values, which correspond to more peaked DSDs and lower intensity rainfall (less than 5 mm h$^{-1}$).

3. The obtained $\mu$-$\Lambda$ relationships are consistent with other relationships from the literature.

4. The new power-law model looks very similar to previously proposed polynomial models but offers better physical interpretation. For example, Eq. 18 shows how the order of the moments used to fit the DSD data influence the $\mu$-$\Lambda$

relationship.

While this study gives further insight into $\mu$-$\Lambda$ relationships and their differences between stratiform and convective rainfall in the Netherlands, it is still necessary to further investigate the impact of having two clearly different relations during DSD retrievals and whether the correct choice of the relationship matters for a given retrieval algorithm and rainfall intensity. Also, more convective-type events should be considered to get a more representative idea of the natural variability of $\mu - \Lambda$ relations

within and between events. Currently, a new extended DSD dataset is being prepared, which is expected to provide further insights into these issues. Finally, a future work could further investigate the characteristics of the discarded DSDs to determine when the two sensors exhibit the most significant differences and under which rainfall regime.

*Author contributions.* CG mainly worked on data processing, visualization of the results and writing (original draft preparation). MS and CU focused on the supervision of CG with fundamental ideas about the direction of the research, the used methodology and finally the writing 280 (review and editing).

*Competing interests.* The authors declare that they have no conflict of interest

*Acknowledgements.* This work was supported by the "User Support Programme Space Research 2012-2016", project ALW-GO/15-35 as well as the Ruisdael Observatory, a scientific research infrastructure co-financed by the Netherlands Organisation for Scientific Research (NWO), grant number 184.034.015.

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

**Table 1.** Overview of the selected convective events. Date, number of 1-minute samples, mean ($\overline{x}$) and standard deviation ($\sigma$) of rain intensity (R), generalized intercept parameter ($N_w$), mass-weighted mean diameter ($D_m$) and liquid water content (LWC). Note that the number of samples denotes the total number of 1-min samples available after filtering (both disdrometers combined).

| Event | Date | No. of samples | R ($\overline{x}$ / $\sigma$) (mm h$^{-1}$) | $N_w$ ($\overline{x}$ / $\sigma$) (mm$^{-1}$m$^{-3}$) | $D_m$ ($\overline{x}$ / $\sigma$) (mm) | LWC ($\overline{x}$ / $\sigma$) (g m$^{-3}$) |
|---|---|---|---|---|---|---|
| 1 | 17/08/2022 | 10 | 32.4 / 13.5 | 708.3 / 232.9 | 2.8 / 0.7 | 1.4 / 0.5 |
| 2 | 30/06/2022 | 16 | 15.2 / 4.5 | 974.7 / 138.1 | 1.7 / 0.2 | 0.9 / 0.2 |
| 3 | 24/06/2022 | 22 | 66.1 / 33.6 | 2604.7 / 341.2 | 2.4 / 0.4 | 3.5 / 1.6 |
| 4 | 19/05/2022 | 9 | 123.1 / 11.1 | 4460.4 / 597.2 | 2.4 / 0.2 | 6.1 / 0.5 |
| 5 | 05/07/2021 | 19 | 16.0 / 3.5 | 1193.0 / 258.3 | 1.6 / 0.2 | 1.0 / 0.2 |
| 6 | 04/07/2021 | 14 | 15.1 / 4.1 | 443.8 / 62.7 | 2.2 / 0.2 | 0.7 / 0.2 |
| 7 | 03/07/2021 A' | 21 | 18.8 / 8.2 | 982.9 / 222.2 | 1.8 / 0.3 | 1.0 / 0.3 |
| 8 | 03/07/2021 B' | 31 | 20.9 / 5.5 | 898.1 / 458.5 | 2.5 / 0.5 | 1.0 / 0.3 |
| Overall Convective | - | 142 | 30.8 / 29.9 | 1315.5 / 977.5 | 2.2 / 0.5 | 1.6 / 1.5 |
| Overall Stratiform | - | 16833 | 1.8 / 3.9 | 394.3 / 417.4 | 1.2 / 0.4 | 0.2 / 0.4 |

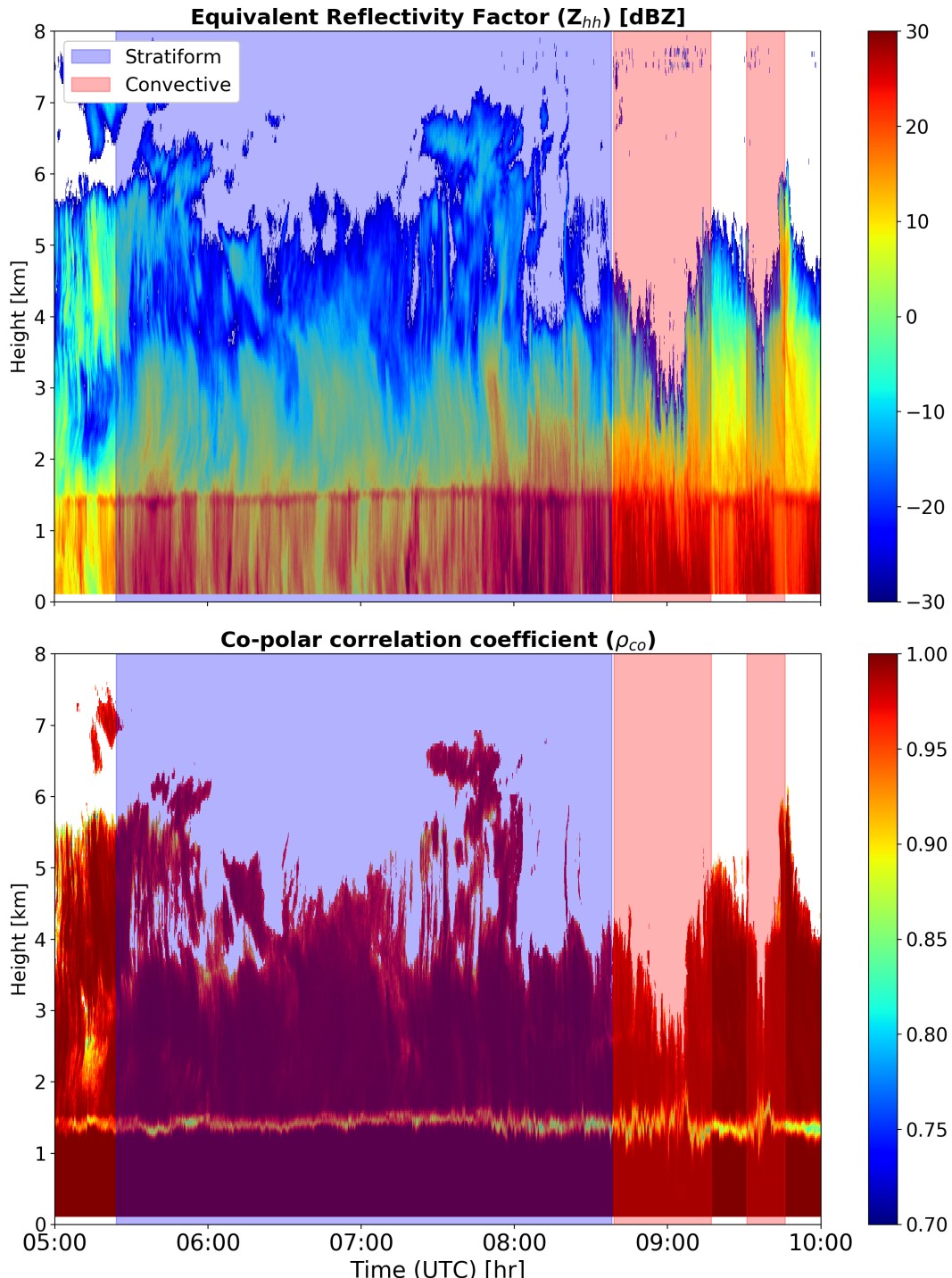

**Figure 1.** Classification of a stratiform event on May $22^{nd}$ 2021 based on the BR03 method. Height–time plots (top to bottom) of reflectivity factor (dBZ) and co-polar correlation coefficient from cloud radar.

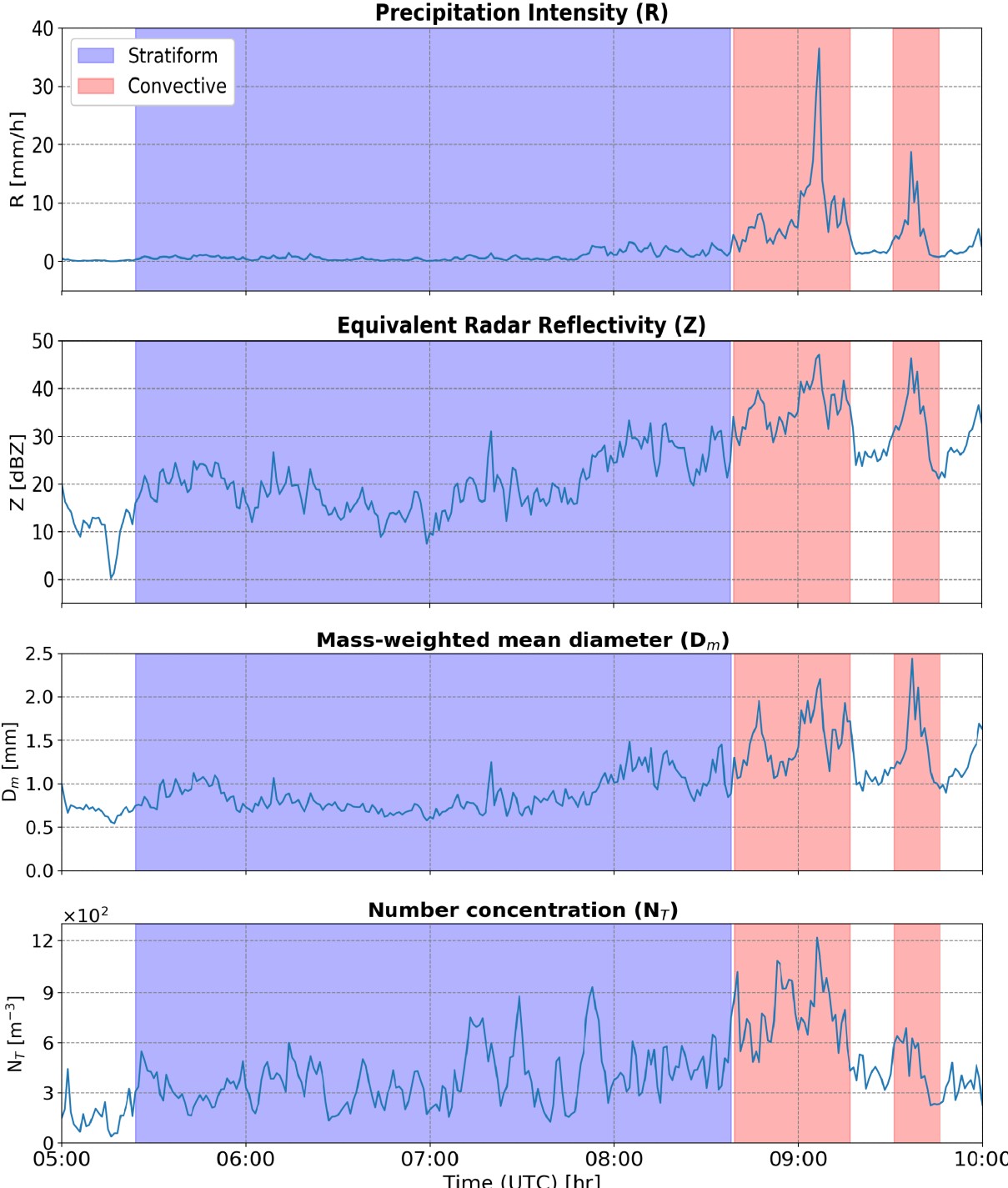

**Figure 2.** Classification of a stratiform event on May $22^{nd}$ 2021 based on the BR03 method. Time series (top to bottom) of precipitation intensity (mm h$^{-1}$), equivalent reflectivity factor (dBZ) in the Rayleigh scattering regime, mass-weighted mean diameter (mm), and number concentration (m$^{-3}$) from Parsivel disdrometer. Note that after 09:00 there is a peak in rainfall intensity that caused strong attenuation of the cloud radar signal.

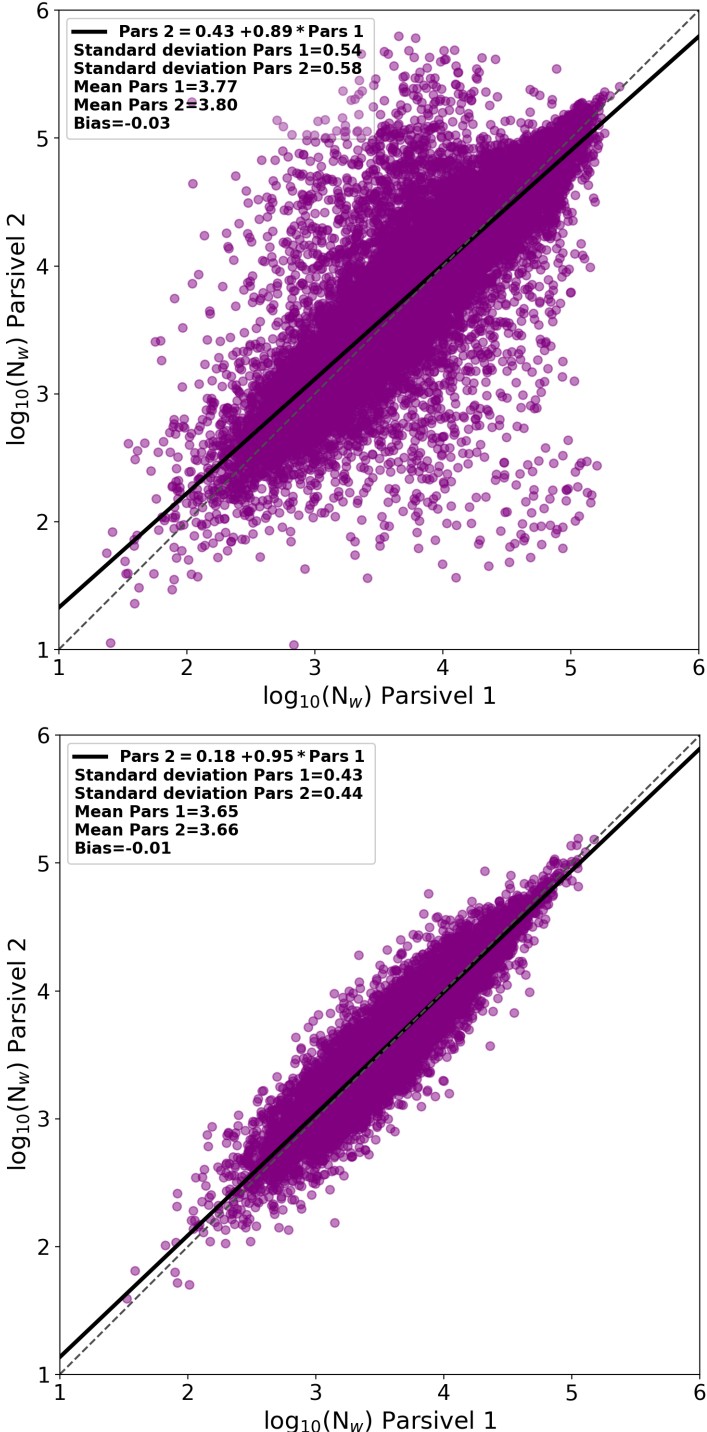

**Figure 3.** Scatter plots of $\log_{10}N_w$ between Parsivel 1 and Parsivel 2 (top to bottom) before and after $D_m$ and LWC quality control filter.

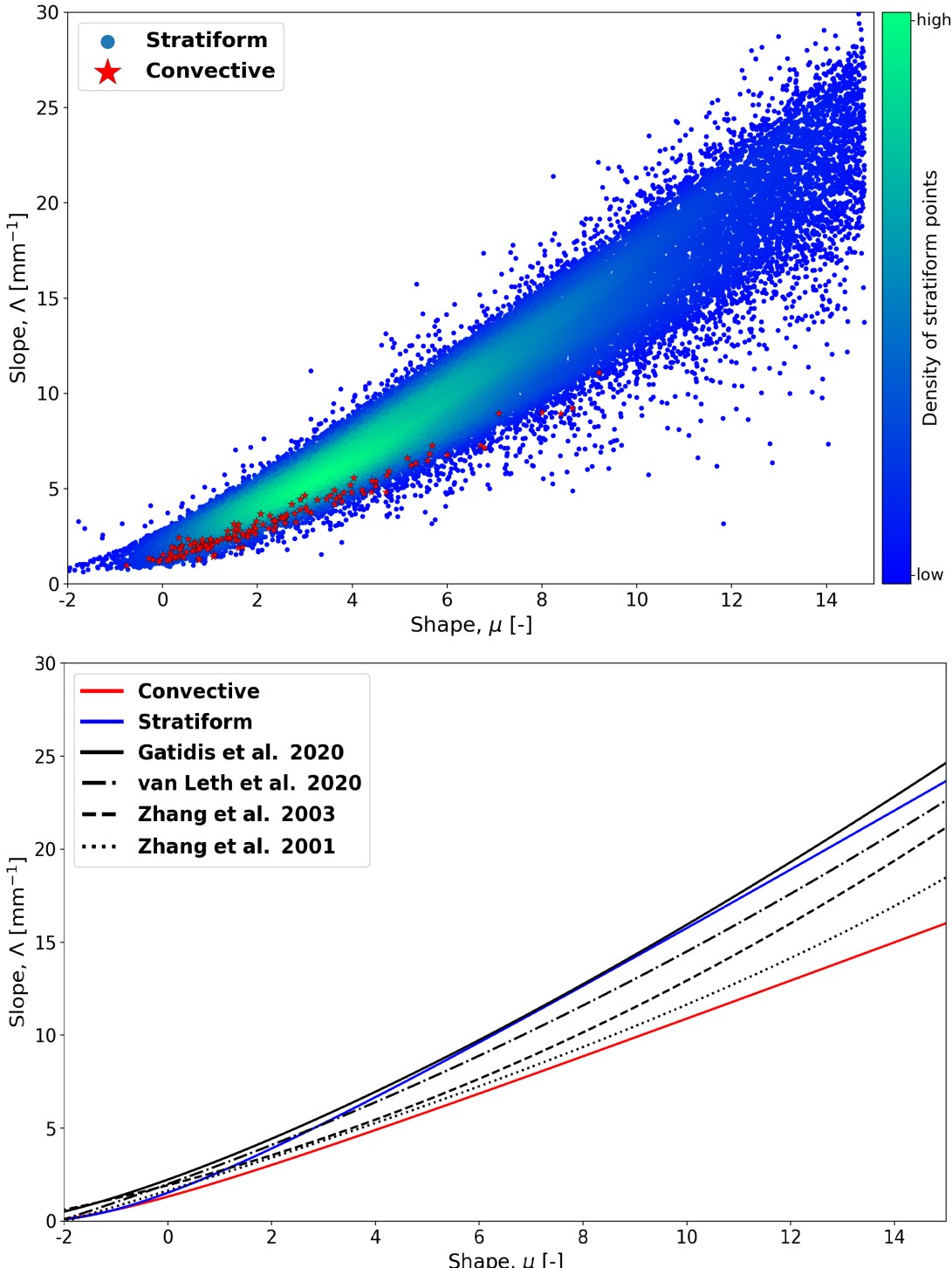

**Figure 4.** Top panel: $\mu-\Lambda$ pairs for convective rain (stars) and stratiform rain (points). The density of stratiform points increases from blue to green. Bottom panel: $\mu-\Lambda$ relationships for convective and stratiform rain types, together with commonly cited models from the literature.