# Peer review of "A new power-law model for $\mu$ - $\Lambda$ relationships in convective and stratiform rainfall"

_Atmospheric Measurement Techniques, 2023_

## Referee Comment (RC2)

REVIEW REPORT

Review of amt-2023-155

By Christos Gatidis, Marc Schleiss, and Christine Unal

Manuscript Title – μ-λ relationships for convective and stratiform rainfall in the Netherlands

**GENERAL COMMENTS**

The Authors proposed a new model to represent the μ-λ relationships. The parameters of the proposed μ-λ relationships are obtained considering 20 months of disdrometers data in the Netherlands. μ-λ relationships for stratiform and convective conditions are also obtained and compared with models in the literature. The manuscript is well written and easy to follow. I suggest the publication on Journal of Hydrology after addressing my comments.

**MAIN COMMENTS**

1) I suggest to slightly change the title in order to stress the fact that in the paper a new model is proposed to model the μ-λ relationships
2) In the Introduction (last sentence) it should be highlighted that a new model is proposed to model the μ-λ relationships and the advantages of this model with respect to the classical ones
3) Section 3.1 To help the reader please add which moments the Authors use to fit the gamma DSD. Furthermore, recent works have criticized Method of Moments for producing biased parameters, whereas the maximum likelihood method proves to perform better (see e.g. Smith and Kliche, 2005 ; Smith et al., 2009 ; Kliche et al., 2008 ). Please provide some comments/consideration on this important aspect.
4) Equation 13: can the Authors write the equation of Mj and Mj-1 that lead to the right hand side?
5) Section 3.3: please add more information on the methodology used to classify stratiform and convective period. The classification is done for each minute or on a longer time period? How the lighting information are used for the classification?
6) Section 4.1: please insert the data quality methodology in the previous section
7) Section 4.1: Is it possible to know if the discarder DSDs correspond mostly to convective of stratiform period? It would be interesting to know some characteristic of the discarded DSDs to understand when the two devices differ more
8) Section 5, point 1): to check this conclusion two μ-λ relationships can be obtained (one for each disdrometer) and then compared.

REFERENCES

Smith, P.L., Kliche, V.D., 2005. The bias in moment estimators for parameters of drop size distribution functions: sampling from exponential distributions. J. Appl. Meteor. 44, 1195–1205. http://dx.doi.org/10.1175/JAM2258.1 .

Smith, P.L., Kliche, D.V., Johnson, R.W., 2009. The bias and error in moment estima- tors for parameters of drop size distribution functions: sampling from gamma distributions. J. Appl. Meteorol. Clim. 48, 2118–2126. http://dx.doi.org/10.1175/ 2009JAMC2114.1 .

Kliche, D.V., Smith, P.L, Johnson, R.W., 2008. L-moment estimators as applied to gamma drop size distributions. J. Appl. Meteorol. Clim . 47, 3117–3130. http://dx. doi.org/10.1175/2008JAMC1936.1 .

---

## Author Comment (AC1)

**Response to the reviewers' comments**
* * *
**Reviewer 1**

*This is an interesting study about convective and stratiform relationships of rain drop size distributions. The interpretation of the mu – gamma relationship parameters is a particularly valid contribution. Overall, I think the manuscript should be published but I recommend complementing some aspects (introduction), reorder (quality control should no be described in the results sections) and fix a few details – please see below.*

Answer

Thank you for taking the time to review our paper and for the valuable suggestions. We carefully revised the initial draft based on the provided input, please see below for more details.

*1)     Page 1, abstract. "In this study, we take a closer look at the important issue of µ-Λ relationships in raindrop size distributions (DSD) by conducting a systematic analysis of twenty months of rainfall data in the Netherlands". I think the abstract should specify that the main analysis is performed with disdrometer data, as other approaches are possible, for example using horizontal radar reflectivity fields or vertical profiles (see next comment).*

Answer

Done. This is now clearly stated in the abstract. We re-formulated the above sentence as follows:

"In this study, we take a closer look at the important issue of µ-Λ relationships in raindrop size distributions (DSD) by conducting a systematic analysis of twenty months of data collected by disdrometers in the Netherlands."

*2)     Page 2, Introduction. As mentioned in the previous comment I recommend to briefly expand the initial part of the introduction with a comment about general approaches to classify convective and stratiform precipitation. Now the first sentence of section 3.3 seems to do this function but I do not think it is complete or well located there. I suggest moving it to the first paragraph of the introduction and comment that other techniques may include radar scanning (polar volumes) radar data or vertical profiles (see for example Qi et al 2013, Powell*

*et al 2016, Ghada et al 2022, Romatschke & Dixon 2022), as well as ground based rain gauge or disdrometer data, as already explained in the paper.*

Answer

As you suggested, we added some additional sentences in the Introduction to highlight this important issue:

"Another issue that arises when studying µ-Λ relationships is the rainfall classification. Several techniques have been proposed to classify rainfall into stratiform and convective regimes using a variety of different sensors. These methods may include weather radar data, Micro Rain Radar (MRR) vertical profiles and machine learning models for the bright band detection (Ghada et al., 2022; Romatschke and Dixon, 2022; Qi et al., 2013; Powell et al., 2016). For example, Yang et al. (2019) used a K-nearest neighbor supervised machine learning algorithm to classify precipitation types. Doppler radar data were used to train the model, with the results to indicate that the algorithm is capable of accurately classifying most of the convective cases and almost all the stratiform ones. Other studies use a combination of ground based sensors like rain gauges or disdrometers and radar data (Ulbrich and Atlas, 2007; Tokay and Short, 1996; Bringi et al., 2003)."

3)  Page 3, Data. The text indicates that two co-located Parsivel2 units are used. Could you please briefly comment if the plane of measurements are aligned? At some sites co-located Parsivel units are installed perpendicularly.

Answer

Yes, we confirm that the two sensors are oriented perpendicularly to each other. We added that detail in the revised manuscript:

"The DSD data used in this study were collected by two co-located, perpendicularly oriented Parsivel$^2$ (Particle Size and Velocity) optical disdrometers (hereafter Parsivel 1 and Parsivel 2) in Cabauw, a polder area located in the western part of the Netherlands between January 1$^{st}$ 2021 and August 31$^{st}$ 2022."

*4)  Page 3, Data. Quality control. I think that an overview of the data quality control should be given either in the Data or Methodology section, and then explain the results in current section 4.1. For example, did author considered some conditions on number of particles present at each 1-min time slot to be considered valid (as in Hachani et al 2017)? Moreover, no mention is made about possible effects of wind on the data (see for example Friedrich et al. 2013, Li et al 2021).*

Answer

Thank you for your comment. The answer is: no, we did not apply any other additional constraints on the number of detected particles, drop sizes or rainfall rates. There is no need to do this, because the cross-check between the two disdrometers

automatically flags all the problematic DSD spectra for which the two sensors do not agree (either due to wind effects or low number densities).

To clarify this issue, we moved part of the description of the quality control steps to the Methodology (see Section 3.3, DSD filtering) and only left the results of the quality control in Section 4.1 (Results).

Regarding your comment on the effect of wind on the data, we added the following part in the manuscript:
 "In the past, several studies have highlighted the effect of strong winds on Parsivel observations (Friedrich et al., 2013a; Lin et al., 2021), which could result in unrealistic big raindrops with small fall velocities. Thus, Friedrich et al. (2013b) proposed a quality control method for removing all these spurious observations. In present work even though no action was taken in this direction, the observations from the two co-located sensors were compared to each other. Whenever the agreement between the two sensors was low, the DSDs were removed from the analysis. ... A detailed description of the filtering process will be given in a following section. No effort was made to investigate the reasons behind the occasional disagreements. The latter have already been extensively studied and documented in the literature and include, among other, errors due to wind, sampling, splashing and internal processing."

*5)    Page 3, Data. How was the ZEUS lightning data used? Did authors check if they were present in some specifi time/range window? In page 7, line 138, the 4 item listed does not specify. Please comment briefly in the Data section or in 3.3. section.*

Answer

Thanks for your comment. We re-formulated the section about the stratiform/convective classification and particularly the part about the lightning data to avoid any confusion to the reader:
"To determine the convective events, we start by identifying all 1-min DSD measurements for which the rain rate exceeds 10 mm/h. We then remove all periods for which there is a clear melting layer signature, since these correspond to stratiform rain. Regarding requirements 3 and 4, please note that no processing was performed on the associated data sets. CAPE and lightning activity are only used as additional diagnostic variables to help with the final classification decision. For the final selection of convective events, only the periods for which the CAPE values were larger than 1000 J/kg and for which lightning strikes were detected over the Cabauw area are kept. ... In this study, they are used as an additional indicator for potential convection which together with the high rain intensity and the absence of melting layer will ensure that no false convective events are identified."

*6)    Page 3, Data. Finally I recommend that the number of data used (total and final valid minutes) is indicated in the Data section.*

Answer

We added this information in the Data section:
"The total dataset used for this study consisted of 21,178 1-minute DSDs. After filtering, the dataset was reduced to 16,975 DSDs."

7) Page 7. As mentioned I suggest to move the description of the QC to the Data section and only, if completely necessary, leave here the results. By the way, note that 'Parsivel 1 and 2' are not properly introduced – this can be easily done in section Data, something as "two co-located Parsivel2 units are used (hereafter Parsivel 1 and Parsivel 2)".

Answer

Thanks for the suggestion. We moved the quality control overview in Methodology section (Section 3.3), following your 4$^{th}$ comment.

Regarding your comment on Parsivel 1 and 2 which were not properly introduced, we added the following sentence in the Data:
"The DSD data used in this study were collected by two co-located, perpendicularly oriented Parsivel$^2$ (Particle Size and Velocity) optical disdrometers (hereafter Parsivel 1 and Parsivel 2) in Cabauw, a polder area located in the western part of the Netherlands between January 1$^{st}$ 2021 and August 31$^{st}$ 2022."

8) *Page 17, Figure 4. Should the convective fit line be restricted to the maximum value of the data used for the fitting? Please comment.*

Answer

Thanks for the suggestion but we do not think that this is a good idea. One reason for fitting a model is to be able to provide predictions for all possible values of $\mu$, including the ones beyond the range of what has been observed. The model is more general than the observations we have. In our case, the largest $\mu$ value in convective events was approximately 9. Obviously, if we use the model outside of that range (i.e., $\mu>9$), we cannot properly assess the accuracy of the predictions. But in the graph, we can always show the model predictions over the whole range of possible mu values.

In order to make this clear, we added the following sentence in the results part (Section 4.2):
"However, it should be highlighted that predictions for $\mu >9$ in convective events should be interpreted very carefully, given that we do not have any observations beyond this range."

Formal

Page 2. Please check format of the references (for example page 2, line 34, etc.). **Done**
Page 8, line 178. I could not find Section III C (why in Roman numbers?) – please correct. **Done**
Page 9. Equation 9. The format of the lower integration limit given (D=0) is not consistent with Equation 10. Please correct. **Done**
Page 16, Figure 3. Could you please produced the scatter plots as square figures (not rectangles), i.e. with the same length in the x-axis and the y-axis? This allows an easier visual comparison, particularly when the magnitudes in both axis are the same. **Done**

---

## Author Comment (AC2)

**Response to the reviewers' comments**
* * *
**Reviewer 2**

*The Authors proposed a new model to represent the µ-λ relationships. The parameters of the proposed µ-λ relationships are obtained considering 20 months of disdrometers data in the Netherlands. µ-λ relationships for stratiform and convective conditions are also obtained and compared with models in the literature. The manuscript is well written and easy to follow. I suggest the publication on Journal of Hydrology after addressing my comments.*

Answer

We appreciate your time and effort in reviewing our paper, as well as your valuable suggestions. We have revised the initial draft basis your feedback, and you can find a point-by-point response to the comments below.

*1)    I suggest to slightly change the title in order to stress the fact that in the paper a new model is proposed to model the µ-λ relationships.*

Answer

Thanks for your comment. We re-formulated the title to highlight the fact that a new power-law model is used for the µ-Λ relationship. The new title is:
"A new power-law model for µ-Λ relationships in convective and stratiform rainfall".

*2)   In the Introduction (last sentence) it should be highlighted that a new model is proposed to model the µ-λ relationships and the advantages of this model with respect to the classical ones*

Answer

Following your advice, we re-formulated the last part of the Introduction and now the advantages of the new model are clearly highlighted.
 "Within the double-moment normalization framework, a new µ-Λ power-law relationship is introduced and fitted to the remaining data, resulting in coefficients with meaningful physical interpretation.".

*3) Section 3.1 To help the reader please add which moments the Authors use to fit the gamma DSD. Furthermore, recent works have criticized Method of Moments for producing biased parameters, whereas the maximum likelihood method proves to perform better (see e.g. Smith and Kliche, 2005 ; Smith et al., 2009 ; Kliche et al., 2008 ). Please provide some comments/consideration on this important aspect.*

Answer

Done.

The revised text:
"Similarly to Bringi and Chandrasekar (2001); Gatidis et al. (2020); Thurai et al. (2014), the method of moments and more particularly the $3^{rd}$ and $4^{th}$ DSD moments were used to fit the gamma DSD and estimate the three unknown parameters μ, Λ and $N_w$ from empirical DSD spectra, with μ values ranging between -3 and 15, as described by Thurai et al. (2014).

About the second comment: Yes, we are well aware of the limitations of Method of Moments (MoM). In fact, in a previous study of ours (Gatidis et al., 2020), we investigated this very important issue by comparing MoM and Maximum Likelihood Estimation (MLE). Our conclusion was that the performance of the different fitting procedures depends on the characteristic of the DSD spectra themselves and the accuracy of disdrometer measurements (e.g., the ability to correctly capture small and large drops). For example, MLE performs better than MoM if the goal is to fit low order moments such as NT. Also, MLE is superior to MoM for cases where the DSD really follows a gamma distribution. On the other hand, when the true distribution deviates from the gamma, or when the goal is to accurately reproduce specific, higher order moments such as LWC, or Z, MoM tends to be the superior choice.

Gatidis, C., Schleiss, M., Unal, C., and Russchenberg, H.: A Critical Evaluation of the Adequacy of the Gamma Model for Representing Raindrop Size Distributions, Journal of Atmospheric and Oceanic Technology.

In order to make this clear in the paper, we added the following sentence:
"The advantages and disadvantages of method of moments with respect to other methods such as maximum likelihood estimation were discussed in previous studies (Smith and Kliche, 2005; Smith et al., 2009; Kliche et al., 2008; Gatidis et al., 2020) and will not be repeated here."

*4)   Equation 13: can the Authors write the equation of Mj and Mj-1 that lead to the right hand side?*

Answer

Thanks for your comment. We re-formulated the text in order to avoid any confusion to the reader. The new sentence in the manuscript is the following:

"Considering that the DSD is assumed to follow a gamma model, and given that $\int_0^\infty D^a e^{-bD} dD = \Gamma(\alpha+1)/b^{(\alpha+1)}$ and $\Gamma(\alpha+1) = \alpha\Gamma(\alpha)$, where $\Gamma(\alpha)$ is gamma function, then $D_c$ (the ratio of two successive reference moments with i=j-1) is given by:"

5) *Section 3.3: please add more information on the methodology used to classify stratiform and convective period. The classification is done for each minute or on a longer time period? How the lighting information are used for the classification?*

Answer

We added some additional information about this in the manuscript (Section 3.4):
"To determine the convective events, we start by identifying all 1-min DSD measurements for which the rain rate exceeds 10 mm/h. We then remove all periods for which there is a clear melting layer signature, since these correspond to stratiform rain. Regarding requirements 3 and 4, please note that no processing was performed on the associated data sets. CAPE and lightning activity are only used as additional diagnostic variables to help with the final classification decision. For the final selection of convective events, only the periods for which the CAPE values were larger than 1000 J/kg and for which lightning strikes were detected over the Cabauw area are kept. High CAPE level indicate favourable conditions for strong updrafts and storm development, potentially leading to convective rain, while lightning is a phenomenon that can accompany convective storms. However, it is important to state that they are not the exclusive drivers of convective processes (Schumacher et al., 2013). In this study, they are used as an additional indicator for potential convection which together with the high rain intensity and the absence of melting layer will ensure that no false convective events are identified. The reasoning behind this approach is that we think it is preferable to be too strict and exclude a few convective events rather than being too tolerant and include some stratiform or mixed-type events into the convective dataset."

6) *Section 4.1: please insert the data quality methodology in the previous section*

Answer

Thank you for your comment. We moved the overview of the quality control in Methodology (see Section 3.3, DSD filtering) and only left the results of the quality control in Section 4.1 (Results).

7) *Section 4.1: Is it possible to know if the discarder DSDs correspond mostly to convective of stratiform period? It would be interesting to know some characteristic of the discarded DSDs to understand when the two devices differ more*

Answer

A further investigation of the characteristics of the discarded DSDs would be an interesting follow-up study. However, we feel that this is beyond the scope of this study. Also, a systematic analysis of discarded DSDs would require a different procedure for rain type classification. As stated in Section 3.4, the current method for identifying convective events is based on a few simple quantitative criteria (e.g., rain rate above 10 mm/h), together with a more subjective and qualitative visual analysis by human experts (i.e., melting layer detection from MRR and cloud radar, CAPE and lightning activity). In the manuscript, we clearly state that these criteria are quite strict, which means that we may have missed some convective events. Therefore, if we wanted to analyze the properties of the discarded DSDs in more detail, we would need a more elaborate, reliable and automatic rain-type classification method.

We added the following sentence at the end of the revised manuscript as an interesting follow-up study:
"Finally, a future work could further investigate the characteristics of the discarded DSDs to determine when the two sensors exhibit the most significant differences and under which rainfall regime."

8) *Section 5, point 1): to check this conclusion two μ-λ relationships can be obtained (one for each disdrometer) and then compared.*

Answer

Thank you for your suggestion. The μ-Λ relationship for each disdrometer was obtained and then compared. There is relatively good agreement between the sensors. We added the following sentence in the revised manuscript:
"Furthermore, the μ – Λ relationship for each disdrometer was obtained and then compared, with a relatively good agreement between the two sensors, especially for smaller μ values (μ < 4) where RMSD is 0.28 while for cases with μ greater than 4 RMSD is increased to 1.1. The slightly bigger differences between the two relations for higher μ values can be explained by the existing sampling uncertainty in the lower rainfall intensities. All the above imply that a single disdrometer may suffice to derive representative μ-Λ relationships without requiring co-location."